# Competency Model for the Middle Nurse Manager (MCGE-Logistic Level)

**DOI:** 10.3390/ijerph18083898

**Published:** 2021-04-08

**Authors:** Alberto González-García, Arrate Pinto-Carral, Jesús Sanz Villorejo, Pilar Marqués-Sánchez

**Affiliations:** 1Department of Nursing and Physiotherapy, Leon University, 24071 León, Spain; agong@unileon.es (A.G.-G.); pilar.marques@unileon.es (P.M.-S.); 2University Dental Clinic, European University of Madrid, 28045 Madrid, Spain; jesus.sanz@universidadeuropea.es

**Keywords:** model of competencies, competency, middle nurse manager, nurse manager, logistic level, health policy, healthcare services, healthcare affordability, Hospital efficiency

## Abstract

Healthcare systems are immersed in transformative processes, influenced by economic changes, together with social and health instability. The middle nurse manager plays a fundamental role, since he or she is responsible for translating the strategic vision, values and objectives of the organization. The objective of this study was to propose the model of competencies to be developed by the middle nurse manager in the Spanish healthcare system. Our methodology consisted in the application of the Delphi method in order to reach an agreement on the necessary competencies, and principal component analysis (PCA) was used to determine the construct validity, reducing the dimensionality of the set of data. Fifty-one competencies were identified for the definition of the model, highlighting decision-making, leadership and communication. The PCA pointed out the structural validity of the proposed model through the saturation of the main components (α Cronbach > 0.631). The results show the model of competencies which the middle nurse manager in the Spanish healthcare system must develop. Middle nurse managers may use these as criteria to plan their professional strategies in the context of management. This model of competencies can be applied to establishing selection processes or training programs for the role of middle nurse manager.

## 1. Introduction

Healthcare systems are immersed in processes of global transformation, influenced by economic and social changes, changes in health technology and structural alterations in systems for the provision of healthcare [1,2,3,4,5]. In this uncertain context, nurses are under pressure to improve quality of care [6]. It thus seems logical that nurses should form part of the nucleus of healthcare so that organizations are able to deal with these changes successfully [7,8,9]. Thus, as highlighted by Witt et al. [10], when the nurse takes part in the healthcare process (management and nursing care) organizations achieve better performances [11,12]. Within health organizations, nurse managers are a key part of any healthcare attention team. Nurse managers are responsible for introducing changes and creating environments in which nurses are able to provide quality attention, at the same time as guaranteeing the achievement of the objectives of the organization under sustainability and efficiency criteria [13,14,15].

The relationship between economic and sustainability policies with respect to offering quality care in health systems is the starting point and is of interest in justifying the development of managerial competencies, which are related to a higher degree of performance and results [16,17,18,19]. In this sense, Yoder-Wise [20] states that that the development of an advanced level of managerial competency is fundamental in achieving the objectives of the organization. Warshawsky [21] highlights that one of the key strategies for the success of health organizations currently resides in the capacity of the nurse manager to develop advanced management skills. This development is achieved through carrying out postgraduate university studies [22]. In this sense, West [23] states that it is possible to observe a difference between nurse managers who have undertaken university programs in management compared to others who have not completed this type of training program. For his part, Herrin [24] points out how master’s degree training empowers nurse managers, enabling the effective management of the healthcare process.

Therefore, in order to identify, orient and train nurse managers, managerial competencies are an essential resource [25]. This competency training in management must go beyond the ambit of nursing, for example, including business management, artificial intelligence, technology, etc. [26,27,28].

Although there is no single definition of management competency [29], we can define it as the correct combination and application of the knowledge, attitudes and skills of middle nurse managers in specific management functions, which are observed and measured as behaviors [30]. New [31] defines managerial competencies as those in which the nurses are able to collaborate with other people, whereast Hudak et al. [32] define them as the skills, knowledge and capacities necessary to achieve quality healthcare.

In healthcare organizations, there is a chain of authority from upper management to the assistance level (Figure 1) [33]. A middle nurse manager (known in Spain as a “supervisora de área de enfermería” or “jefa de área de enfermería”) is the person in the intermediate position between the operational level of the nurse manager and the nurse executive [34]. The middle nurse manager is responsible for translating the culture and strategy of the organization to the operational level, as well as managing resources, coordinating nursing care and planning and contributing to the evaluation of services provided, together with supporting and encouraging teamwork in the attention units and implementing innovative practices [33,35,36,37,38]. Therefore, the middle nurse manager plays a key position, since they do not only carry out clinical leadership and management, but are also responsible for translating the strategic vision and the values and objectives of the organization’s care actions [39,40]. In the Spanish healthcare system, the middle nurse manager is responsible for the management and coordination of a functional area of nursing in a healthcare organization, for example, the surgical area [41].

Managerial competencies have been researched from various angles. For example, Chase [41] identified technical, human and conceptual skills, as well as leadership and financial management skills. The American Organization of Nurse Executives (AONE) [36] identified the management of the relationships, communication, leadership, knowledge of the health environment and financial skills as strategic areas in the development of competencies. González García et al. [42] highlighted the management of relationships, communication skills, listening, leadership, conflict management, ethical principles and skills for managing teams as core competencies for nurse managers. Finally, in a brief summary, Pillay [43] highlighted the management of people and organizational capacity, together with strategic thinking, as key competencies.

On the other hand, based on the literature review, it can be deduced that it is necessary to improve the state of knowledge about the role of the middle nurse manager [35,44,45]. The competencies necessary are not usually clearly defined, which would explain this gap in the understanding of the middle nurse managers. This same absence is evident in the Spanish context, as there is no model of competencies for the carrying out of management functions at the logistical level.

For this reason, the main objective of this study was to propose a model of competencies which should be developed by a middle nurse manager in the Spanish healthcare system. For this reason, the following specific objectives were proposed:Reach a consensus on the competencies required for a middle nurse manager.Establish a consensus on the degree of development of each of the competencies required for a middle nurse manager.Achieve consensus about the training required to develop each competency.Assess the structural validity of the proposed model.

## 2. Materials and Methods

### 2.1. Revision of the Literature

Based on the scoping review [46] of the literature carried out during 2010–2019 to determine the competencies associated with nursing management, electronic databases were used (Web of Science, Scopus, PubMed and CINAHL) to carry out the search, identifying 56 competencies for middle nurse managers. The results of this review provided the basis for carrying out this Delphi study, evaluating the competencies of the positions of the middle nurse manager.

### 2.2. Delphi Methodology

A four-round Delphi method was employed. The Delphi method focuses on the identification of expert opinion to reach a consensus [47]. The Delphi methodology has been considered the most convenient method when there is a lack of knowledge on a topic. [47,48].

The aim of the first Delphi round was to generate a list of competencies required for middle nurse managers. In the second round, experts were provided with feedback from the first iteration. All participants were invited to reconsider their opinion. In the third round, the experts were asked their opinion on the competencies of the middle nurse manager, in order to reach a consensus. The expert panel also was asked to indicate the training necessary to achieve specific levels of competency (expert, very competent, competent, advanced novice and novice). During round four, experts were provided with feedback from the third round and they were encouraged to rethink their original answers after reviewing the report of the third Delphi round.

#### 2.2.1. Consensus

For this research, consensus was set at 80% or greater agreement (defined as somewhat in agreement–total agreement [score 4–5]) with regard to (I) the proposed competencies; (II) the level of development of the competencies; (III) the training for each competency. Items with less than an 80% response rate were eliminated for the following Delphi round.

#### 2.2.2. Participants

Two categories of experts, consisting of a combined total of 50 experts (Table 1), were established based on the Delphi Technique:

Experts in healthcare management. This category of experts represented healthcare from the different hospitals and institutions, and performed management and leadership functions.

Experts in the health environment. This category of experts represented the different fields involved in healthcare and were selected for their specific views on healthcare practices, university training, students and healthcare research.

#### 2.2.3. Variables

The following variables were used:

Sociodemographic variables: to determine the characteristics of the participants, information was collected on age, gender, university degree, university training, postgraduate training, professional role, location of the study, years of national activity, years of management experience, management functions carried out and international experience. Sociodemographic variables were used to establish the profile of the expert panel.

Competencies: the competencies suggested to the experts came from the literature review. The competencies were used to establish the competency model for the middle nurse manager.

#### 2.2.4. Delphi Surveys

Two surveys were developed for the specific purpose of the study.

Competencies necessary for middle nurse managers: Every participant quantified their level of agreement or disagreement with each competence in accordance with the Likert scale from 1 to 5 (1 = disagree totally, 5 = completely in agreement).

Degree of development of the competencies of middle nurse managers: In order to achieve an agreement on the competencies required for each functional level of nurse manager, the level of consensus with each competency was registered in accordance with the Likert scale from 1 to 5 (1 = beginner, 5 = expert), and the type of training required to develop the competencies, in accordance with the Likert scale of 1 to 6 (1 = university extension, 2 = continuous training, 3 = university expert, 4 = diploma in university specialization, 5 = master’s degree, 6 = PhD).

#### 2.2.5. Degree of Development

The term “degree of development” was used to indicate the degree of proficiency shown by middle nurse managers in the performance of each competency. Based on Bernner’s theory [49], therefore, the level of development was expressed as follows.

Novice: a middle nurse manager who has no prior experience in a competency associated with a professional role or situation. In many instances, this is the starting point for a nurse manager, as they would be in possession of clinical competencies, yet lack knowledge and skills in management.Advanced novice: someone who is able to contribute partial solutions to unknown or complex situations. Although an advanced novice may be able to perform the functions required for the nurse manager position, they may or may not have the ability to understand the context and actions required.Competent: implies an adequate understanding of the context and situation. The competent middle nurse manager may be able to cope with situations associated with the nurse executive role, although they may lack analytical skills and an understanding of complex situations.Very competent: the middle nurse manager focuses on a comprehensive understanding of situations at every level, and is someone who is able to anticipate problems and make appropriate decisions.Expert one who demonstrates the behavior of the model of competencies. The expert nurse manager anticipates problems, understands them at an instinctive level and proposes correct and appropriate solutions [49].

The degree of development must be understood in a progressive and exclusive manner, starting at the level of novice and ending at the level of expert. The degree of development is achieved through assigning the appropriate training to each level.

### 2.3. Principal Component Analysis

Principal component analysis (PCA) is a technique for the transformation of data [50]. The main purpose is to reduce the dimensionality of a data set by reducing the number of variables and preserving as much relevant information as possible [50]. Factor analyses were performed according to Thurstone’s theory [51,52] (3 phases): first, determining if the data are suitable for factor analysis; second, performing the extraction of the factors and, finally, carrying out the rotation and interpretation of the factors.

The Kaiser–Meyer–Olkin (KMO) method was used to determine the suitability of the data for the factorial analysis. The following step was the extraction of the data, using the Kaiser criterion, making the decision based on values higher than 1 [53], and a scree plot, which is a graphical representation transit value [54]. Finally, the rotation interpretation of the factors was made by means of the varimax rotation method and Kaiser standardization to obtain the simplest possible structure that was easy to interpret.

## 3. Results

### 3.1. Demographic Data of the Panel of Experts

Fifty experts responded to our invitation to participate and take part in the Delphi study. All of them completed the questionnaires of the Delphi study (100% response rate). The characteristics of the full 50-member Delphi panel are listed in Table 1.

### 3.2. Model of Competencies for the Middle Nurse Manager

During the first and second Delphi rounds, a consensus was reached (more than 80%) for 51 competencies from the proposed list. The consensus of round 2 details the competencies that make up the model (Table 2) structured into six dimensions, according to the following definitive characteristics: management, communication and technology, leadership and teamwork, knowledge of the healthcare system, nursing knowledge and personality (Figure 2). In round 2, competencies with a consensus of less than 80% were eliminated.

The third and fourth Delphi rounds demonstrated that 51 competencies are necessary for middle nurse managers, with a consensus that the development of these competencies should be at the level of “Expert”, “Very competent” and “Competent”. A consensus was also reached during rounds 3 and 4 to achieve the level of development of each competency. The final consensus as detailed in Table 2 and makes up the competency model for the middle nurse manager in the Spanish healthcare system.

As regards the training necessary for the development of each of the levels of competency, in Table 3, we can observe the consensus reached for each of the levels of development of the competency.

### 3.3. Principal Component Analysis

For the principal component analysis, the competencies were grouped together in dimensions, in accordance with their definitive characteristics. The dimensions of management, communication and technology and leadership and teamwork made up four principal components, the dimensions of knowledge of the healthcare system and the personality dimension comprised two principal components, whereas the nursing knowledge dimension was designated as a single main component (Table 4). The factorial loads of each of the items integrated into each dimension widely exceeded the lower level of 0.4, and the α Cronbach demonstrates the quality of the adjustment (Table 4). From these results it can be deduced that the proposed model is structurally sound.

## 4. Discussion

In this study, a competency model for the role of middle nurse manager was developed and validated (In Spain, the middle nurse manager is known as the “supervisora de área de enfermería” or “jefa de área de enfermería”) in the context of the Spanish health system. The model of competencies for middle nurse managers in Spain is made up of 51 competencies, structured into six dimensions, according to their defining characteristics—(1) Management, (2) Communication and Technology, (3) Leadership and Teamwork, (4) Knowledge of the Healthcare System, (5) Nursing Knowledge and (6) Personality. These findings are in line with the arguments of McCarthy and Fitzpatrick [55], who state that the competencies of the middle nurse manager should be oriented towards negotiation, the coordination of resources, the monitoring of the activity, negotiation and empowerment. The AONE in its models suggested the need to develop 35 competencies [56], and Pillay described 51 competencies, building on the research carried out by AONE [43]. However, these models differ from our proposal in terms of the weight that has been awarded to some of the competencies that coincide within the models, such as, for example, the relationships with the management of the business.

During the third and fourth Delphi rounds, agreement was achieved among our experts on the development of the competencies. Agreement was reached for the following levels: “competent” levels (this level is achieved when there is a robust demonstration of competency), “very competent” (this level is considered to be achieved when there is a meaningful demonstration of competency) and “expert” (this level is considered to be achieved when the knowledge and skills of the competency model are demonstrated). This proposal coincides with that of AONE, which uses the levels of competent, proficient and expert for the development of competencies, highlighting how these levels are achieved by means of a master’s degree or PhD studies [57,58]. For the assessment of its relevance, we must highlight that the studies carried out by Chase [40] are different in some ways compared to our research, in that the levels of development of the competencies are not indicated, centering on the degree (minimally, moderately, significantly and essentially for management) to which they contribute to the role of a nursing manager. The results of our research highlight the importance of the strong development of competencies, in the same manner as Crawford et al. [58] emphasized the need for a high level of expertise and the development of a set of competencies to cope with the functions of a logistics-level manager.

During rounds 3 and 4 of the Delphi method, the panel of experts reached a consensus on the training that the nurse manager should develop at the logistic level in three levels of competency (“expert”, “very competent” and “competent”). The “competent” level is achieved by completing continuing education, university expertise and a diploma in university specialization. As regards the “very competent” level, the experts agreed that this is achieved with university expertise, a diploma in university specialization or a master’s degree. Finally, the “expert” level is reached by completing a master’s degree program and PhD studies. It must be borne in mind that work experience does not prepare the nurse to assume management functions, with training being the factor that most significantly influences the development of the competencies of middle nurse managers [59]. Warshawsky et al. [21] warn of the risks to the organization if the nurses assume management responsibilities without the suitable knowledge and training. In the same way, Rizani et al. [60] and Herrin et al. [24] point out that that the competency of the nurse manager is greater when they have carried out advanced studies (a master’s degree or PhD), increasing their level of competency over time to a higher degree than that of the managers who have not carried out advanced training. In this sense, the American Nurses Credentialing Center has made adjustments to the standards of training recommended for all nurse managers, elevating the degree of exigency [61].

The PCA verified the model of competencies for the middle nurse manager, highlighting that the importance of competencies can be defined by three principal components—communication (communication skills, relationship management, conflict management), leadership (leadership skills and team management) and decision-making (decision-making and ethical principles). The eigenvalues demonstrated that the decision-making and ethical principles indicate a strong and significant relationship between these competencies [62]. Furthermore, the eigenvalues also point to the relationship between leadership and work teams [63], and between communication and conflict resolution.

The development of communication skills expected from the middle nurse manager must include the ability to provide critical thinking and stimulate reflection before taking action in nursing teams [35]. It should also provide, for instance, conflict resolution and shared decision-making, which is also associated with team management [64].

## 5. Conclusions

In this study, consensus was reached on the competencies necessary for establishing a model of competencies for middle nurse managers (MCGE-logistic level) (In Spain: “supervisora de área de enfermería” or “jefa de área de enfermería”) adapted to current health policies, economic necessities and the sustainability of the organizations and healthcare in an uncertain environment. In conclusion, this study developed a consensus on 51 competencies necessary for middle nurse managers in Spain, of which the following can be highlighted: communication, leadership and decision-making. The middle nurse manager is accountable for one of the most critical divisions of a healthcare organization, and is essential in the management of nurses and material resources. The quality of the final care provided will depend on their management style. Therefore, a nurse or a nurse manager should not be promoted to the role of middle nurse manager without undertaking advanced programs in management.

The results of our research show the accurate levels of development for each competency for a middle nurse manager. It would be recommendable for the nurse to achieve these competencies before performing the functions of a middle nurse manager.

Any nurse who aspires to carry out the role of a middle nurse manager would be advised to develop the competencies that are set out in the proposed model beforehand.

Furthermore, this study sets out the training necessary to acquire the development of the competencies necessary for the logistical level. Both nurses who wish to be promoted to middle nurse managers and nurse managers who presently work at this level would be advised to follow the education programs that have emerged from our research, in order to adapt their knowledge to the requirements of this role.

### Implications for Nursing Management

This model has implications for the Spanish healthcare system, healthcare policies, and for the practices and education related to middle nurse management. The proposed model contributes to the design of the function of the middle nurse manager, to the selection processes and the design of the study plans of the nurse managers in traditional academic institutions and in programs for continuous professional development within organizations. It is probable that a greater understanding of these competencies can serve as the basis for developing interventions, which could improve the working environment of nurses and patient care, as well as ensuring the safety and the productivity of the organization.

## Figures and Tables

**Figure 1 ijerph-18-03898-f001:**
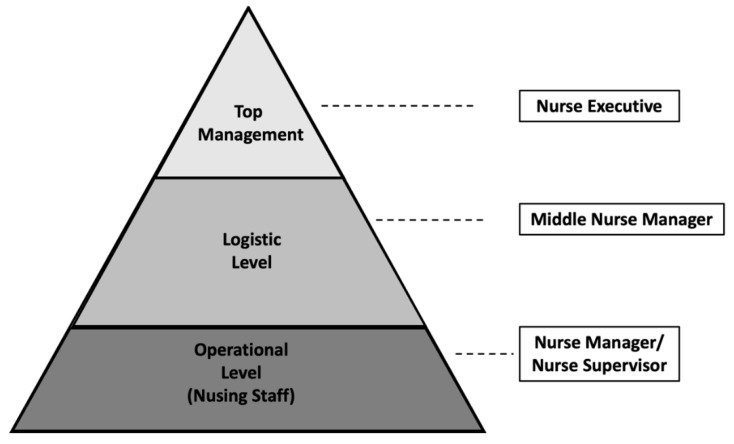
Management levels. Source: own elaboration.

**Figure 2 ijerph-18-03898-f002:**
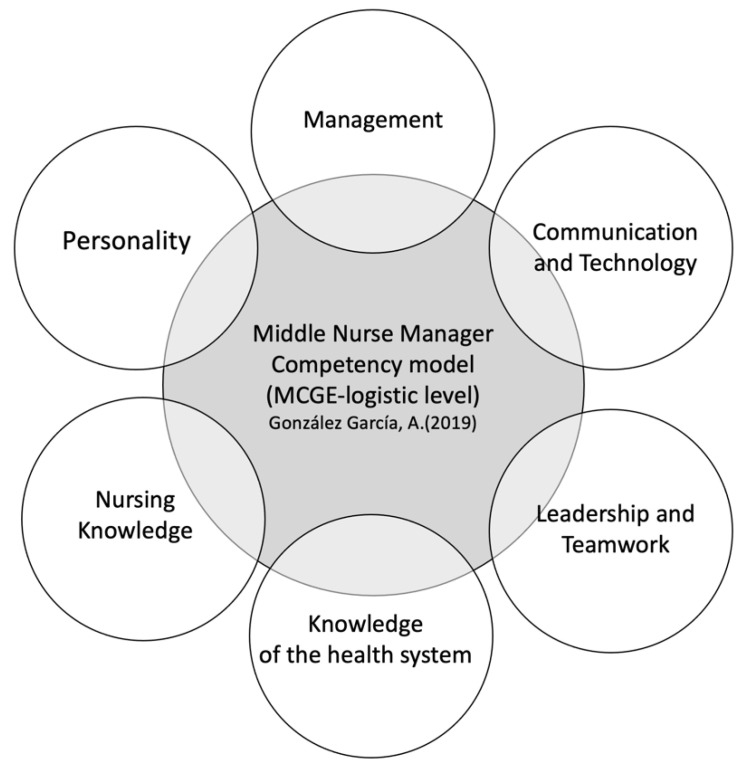
Dimensions of the middle nurse manager executive competency model. Source: own elaboration.

**Table 1 ijerph-18-03898-t001:** Sociodemographic characteristics of the panel experts.

Characteristics	Range/Category	Frequency	Percentage
Age	<40	10	20
	40–50	15	30
	51–60	18	36
	>60	7	14
Sex	Female	32	64
	Male	18	36
Education	Master’s degree	34	68
	Ph.D	14	28
Expert group 1	Minister of Health	3	6.1
Expert group 2	Head of the Health Department	5	10
Expert group 3	General Council of Nurses	3	6
Expert group 4	Scientific Association	4	8
Expert group 5	Trade Union	3	6
Expert group 6	General Manager	5	10
Expert group 6	Medical Director	2	4
Expert group 6	Nurse Executive	5	10
Expert group 6	Management Director	1	2
Expert group 7	Middle Nurse manager	2	4.1
Expert group 8	Nursing supervisor	3	6.1
Expert group 9	Nurse	3	6.1
Expert group 9	Doctor	2	4.1
Expert group 9	Assistant Nursing Care Technician	2	4.1
Expert group 10	Nursing Degree Students	2	4.1
Expert group 11	Research/Teaching	4	8.2
Expert group 12	Lawyer	1	1

Source: own elaboration.

**Table 2 ijerph-18-03898-t002:** Model of competencies for a middle nurse manager.

**I. Management** 1. Analytical thinking (V. COMP) 2. Decision-making (V. COMP) 3. Innovation (V. COMP) 4.Strategic management (V. COMP) 5. Human resources management (V. COMP) 6. Legal aspects (V. COMP) 7. Organizational management (COMP) 8. Result orientation (V. COMP) **II. Communication and technology** 9. Communication skills (V. COMP) 10. Feedback (V. COMP) 11. Evaluation of information and its sources (V. COMP) 12 Listening (V. COMP) 13. Information systems and computers (EXP) 14. Technology (COMP) 15. English medium level of writing (COMP) **III. Leadership and teamwork** 16. Relationship management (V. COMP) 17. Leadership (COMP) 18. Career planning (V. COMP) 19. Influence (V. COMP) 20. Change management (V. COMP) 21. Delegation (V. COMP) 22. Conflict management (V. COMP) 23. Ethical principles (V. COMP) 24. Power and empowerment (V. COMP) 25. Critical thinking (EXP) 26. Collaboration and team management skills (V. COMP) 27. Interpersonal relations (EXP) 28. Multi-professional management (V. COMP) 29. Team-building strategies (V. COMP) **30. Talent management (COMP)**	**IV. Knowledge of the healthcare system** 31. Care management systems (V. COMP) 32. User care skills (V. COMP) 33. Health policy (COMP) 34. Identification and responsibility with organization (V. COMP) 35. Knowledge of the health environment (V. COMP) 36. Quality and safety (V. COMP) 37. Quality and improvement processes (V. COMP) **V. Nursing knowledge** 38. Clinical skills (V. COMP) 39. Standard Nursing Practice (COMP) 40. Nurse Research (COMP) 41. Nursing Theories (COMP) 42. Care Planning (COMP) 43. Nursing training planning (V. COMP) 44. Professionalism (COMP) **VI. Personality** 45. Serve as a model (V. COMP) 46. Awareness of personal strengths and weaknesses (EXP) 47. Strategic vision (V. COMP) 48. Personal and professional balance (V. COMP) 49. Compassion (V. COMP) 50. Emotional intelligence (V. COMP) 51. Integrity (EXP)

Source: own elaboration. Legend of the table: EXP = expert. V. COMP = very competent. COMP = competent.

**Table 3 ijerph-18-03898-t003:** Level of competency development and training.

	Univ. Ext.	Cont. Ed	Univ. Exp.	Univ. Spec. D	Master	Ph.D.
**Novice**	100%					
**Novice Advanced**	90%	98%				
**Competent**		90%	90%	96%		
**Very Competent**		96%	100%	96%	96%	
**Expert**					96%	96%

Source: own elaboration. Note: Univ. Ext. = university extension diploma; Cont. Ed = continuing education; Univ. Exp. = university expert; Univ. Spec. D. = university specialization diploma; Master = master’s degree; Ph.D. = Ph.D. degree.

**Table 4 ijerph-18-03898-t004:** Factor structure of the proposed competency model.

Management Dimension					
	CP1	CP2	CP3	CP4	
Result orientation	0.789				
Strategic management	0.725				
Innovation	0.710				
Legal aspects		0.936			
Analytical thinking		0.554			
Organizational management			0.968		
Decision-making				0.980	
Explained variance	32.325%	18.075%	12.822%	11.569%	
Eigenvalue	2.263	1.265	0.898	0.810	
α Cronbach					0.631
**Communication and Technology Dimension**					
	CP1	CP2	CP3	CP4	
Listening	0.905				
Information systems and computers	0.679				
English medium level of writing		0.874			
Technology		0.636			
Feedback			0.88		
Communication skills			0.589		
Evaluation of information and its sources				0.826	
Explained variance	31.326%	17.341%	16.065%	13.076%	
Eigenvalue	2.193	1.214	1.125	0.915	
α Cronbach					0.6
**Leadership and Teamwork Dimension**					
	CP1	CP2	CP3	CP4	
Change management	0.812				
Influence	0.802				
Leadership	0.703				
Delegation	0.696				
Collaboration and team management skills		0.85			
Critical thinking		0.786			
Team-building strategies		0.736			
Career planning		0.707			
Ethical principles			0.885		
Power and empowerment			0.765		
Conflict management				0.936	
Explained variance	46.309%	12.796%	10.286%	8.17%	
Eigenvalue	5.094	1.408	1.131	0.899	
α Cronbach					0.876
**Knowledge of the Healthcare System**					
	CP1	CP2			
Quality and safety	0.971				
Quality and improvement processes	0.948				
Identification and responsibility with the organization		0.917			
Health policy		0.838			
Explained variance	57.954%	29.970%			
Eigenvalue	2.318	1.199			
α Cronbach					0.749
**Nursing Knowledge**					
	CP1				
Nursing training planning	0.918				
Nurse research	0.910				
Nursing theories	0.822				
Clinical skills	0.777				
Explained variance	73.733%				
Eigenvalue	2.949				
α Cronbach					0.808
**Personality**					
	CP1	CP2			
Awareness of personal strengths and weaknesses	0.905				
Strategic vision	0.891				
Personal and professional balance	0.836				
Compassion		0.884			
Emotional intelligence		0.735			
Explained variance	54.76%	21.705%			
Eigenvalue	2.738	1.085			
α Cronbach					0.809

Source: own elaboration.

## Data Availability

The data presented in this study are available on request from the corresponding author.

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
