# Peer review of "Competency Model for the Middle Nurse Manager (MCGE-Logistic Level)"

_ijerph, 2021, doi:10.3390/ijerph18083898_

Round 1

Reviewer 1 Report

Comments and Suggestions for Authors

Thank you for contributing to this very interesting and current research.

The strength of the paper includes a proper selection of methods, a detailed description of the course of the study.

I consider the following aspects of improvement to be necessary:

  • There is no definition of the Middle Nurse Managers in Spain in the Introduction section it could be completed (short definition).
  • This study aims to reveal the competency of Middle Nurse Managers, and is it not to develop a program for this? Please make this clearer in the introduction.

Detailed comments

  1. Introduction

Page 2, lines 50

Why is it only possible for university research? How is it clinical?

It's hard to understand what it means. Please, clear evidence or explanation is needed.

Page 2, lines 73

What are the administrative duties or their roles or responsibilities for middle nurse managers in Spain? Please explain what is the qualification of the middle nurse manager.

Page 3, lines 91

Previously, what were the competencies of the middle nurse manager in Spain described as? Is it completely absent?

  1. Materials and Methods

Why was the literature reviewed in 20108-2019? Please explain why you chose this period.

2.2.2. Participants

What was the reason you made these two groups of experts(experts in health management and experts in the health environment)? Need a specific explanation of these two groups of experts.

2.2.3. Variables  lines 133

The variables of the study were:

How did you proceed with the research considering these variables?

2.2.5. Degree of development

Are there any standardized standards for this? Please explain the rationale for each of the levels defined.

Page 9

You mentioned

In this research work, a model validating the competencies to be developed by the 222 Middle Nurse Manager (In Spain:” supervisora de área de enfermería”, “jefa de área de 223 enfermería”) in the Spanish Health System is demonstrated.

What does this mean? Please explain more specifically and clearly.

  1. There is no discussion.

Author Response

Response to Reviewer 1 Comments

Title: Competency Model for the Middle Nurse Manager (MCGE-logistic level)

Thank you for reviewing our article and contributing with your feedback to generate an improved version of the paper, which we hope will satisfy the concerns you raised with the previous version. In this new version, changes in the manuscript are highlighted in yellow, so they are easy to identify. We have also included the changes made as a result of each comment as part of this document. Please note that our answers refer to the line numbers in the new version, corresponding to the requests of the reviewers.

We used blue highlight for reviwer 1

We used yellow highlight for reviewer 3

We used red letters for editor

Point 1:

Page 2, lines 50

Why is it only possible for university research? How is it clinical?

It's hard to understand what it means. Please, clear evidence or explanation is needed.

Response 1: Thank you for the suggestion, you can see in lines 50-54 of the manuscript the changes in the way indicated here.

Lines 50-54. We've included it in the manuscript:

In this sense, west (2016) states that it is possible to appreciate the difference between nurse managers who have developed advanced programs in management compared to others who have not completed this type of training programs. For his part, Herrin (2006) points out how master's degree training empowers the nurse manager for effective management of the healthcare process.

Point 2:

Page 2, lines 73

What are the administrative duties or their roles or responsibilities for middle nurse managers in Spain? Please explain what is the qualification of the middle nurse manager.

Response 2: Thank you very much for your appreciation, in lines 76-78 we have included your request

Lines 76 – 78. We've included it in the manuscript:

In the Spanish Healthcare System, the middle nurse manager is responsible for the management and coordination of a functional area of nursing in a healthcare organization, for example, the surgical area.

Point 3:

Page 3, lines 91

Previously, what were the competencies of the middle nurse manager in Spain described as? Is it completely absent?

Response 3: Thank you very much for your appreciation. This research is the first competency model for middle nurse manager in Spain. This is the strength of our research.

Point 4:

  1. Materials and Methods

Why was the literature reviewed in 2010-2019? Please explain why you chose this period.

Response 4: Thank you very much for your appreciation. This is a mistake. It is corrected in the line 107.

Point 5:

2.2.2. Participants

What was the reason you made these two groups of experts (experts in health management and experts in the health environment)? Need a specific explanation of these two groups of experts.

Response 5: Thank you very much for your appreciation. in lines 133-140 we have included your request

Lines 133 – 140. We've included it in the manuscript:

Two categories of experts consisting of a combined total of 50 experts (table 1) were established based on the Delphi Technique:

Experts in healthcare management. This category of experts represented the healthcare from the different hospitals and institutions, and occupied leadership, and management positions.

Experts in the health environment. This category of experts represented the different fields involved in healthcare and were targeted for their specific perspectives on healthcare practice, academic practice, student concerns, research.

Point 6:

2.2.3. Variables lines 133

The variables of the study were

How did you proceed with the research considering these variables?

Response 6: Thank you very much for your appreciation. in lines 147-152 we have included your request

Lines 147 – 152. We've included it in the manuscript:

Sociodemographic variables: In order to define the profile of the expert, information was compiled related to the age, sex, profession, university training, postgraduate training, the professional role, the location of the study, the years of national activity, the years of management experience, the management functions carried out and international experience. Sociodemographic variables were used to establish the profile of the expert panel.

Competencies: the list of the competencies to be proposed for the experts arose from a review of the literature. The competencies were used to establish the competency model for the middle nurse manager.

Point 7:

Page 9

You mentioned

In this research work, a model validating the competencies to be developed by the 222 Middle Nurse Manager (In Spain:” supervisora de área de enfermería”, “jefa de área de 223 enfermería”) in the Spanish Health System is demonstrated.

What does this mean? Please explain more specifically and clearly.

Response 7: Thank you very much for your appreciation. in lines 257-258 we have included your request

Lines 257 – 258. We've included it in the manuscript:

In this research work a model validating the competencies to be developed by the Middle Nurse Manager (In Spain,the middle nurse manager is known as:” supervisora de área de enfermería”, “jefa de área de enfermería”) in the Spanish Health System is demon

Thank you so much for reviewing our paper

Reviewer 2 Report

See comments in attached pdf file.

Author Response

Response to Reviewer 2 Comments

Title: Competency Model for the Middle Nurse Manager (MCGE-logistic level)

Thank you for reviewing our article and contributing with your feedback to generate an improved version of the paper, which we hope will satisfy the concerns you raised with the previous version. In this new version, changes in the manuscript are highlighted in yellow, so they are easy to identify. We have also included the changes made as a result of each comment as part of this document. Please note that our answers refer to the line numbers in the new version, corresponding to the requests of the reviewers.

We used blue highlight for reviwer 1

We used yellow highlight for reviewer 3

We used red letters for editor

The suggestions of reviewer 2 have been applied directly on the text.

Point 1:

File with writing and grammar suggestions

Thank you very much for your contribution. We have considered all of your suggestions regarding grammar and wording of the article. Because of the number of suggestions, we have taken the permission not to point out all the changes on the text. However, if you ask us to do so, we will send you all of them duly marked on the article.

Thank you so much for reviewing our paper

Reviewer 3 Report

The paper sets out to develop a model for competencies needed for successfully manage the logistic level of a nursing environment. The paper uses a Delphi method to identify 51 necessary competencies.   

Major issues

A serious note: how is this paper different from your previously published paper on competencies (Gonzales et al 2020)? It seems to be considerable overlap between these two papers. Please explain how this paper is different and what unique contributions are found in this paper!

Methods: please elaborate on the different levels of development. Are the levels mutually exclusive or are they cumulative? How are we to understand the content of the different levels? Please elaborate!

Discussion: Make an overview of the discussion section and see if you can omit certain sentences and expressions. For instance, you do not need to refer back to the rounds of the Delphi method in the second paragraph. Suggestion: start each paragraph with repeating the specific objective of the paper.

Minor issues

Line 31: Delete unnecessary adjective “enormous”.

Line 95-101: Suggestion, enumerate the specific objectives of the paper.

Line 104: Please add a reference to the term scoping review.

Line 135,140,145,147,155: Delete the bullets for readability.

Table 1: change the interval 41-50 to 40-50 for age.

Table 2: Check the names of the skills (e.g., “Estrategic”, “Lidership”). Categories V and VI should also be in bold face.

Table 4: “Lidership”.

Line 222: Ad a heading for the Discussion section.

Author Response

Response to Reviewer 3 Comments

Title: Competency Model for the Middle Nurse Manager (MCGE-logistic level)

Thank you for reviewing our article and contributing with your feedback to generate an improved version of the paper, which we hope will satisfy the concerns you raised with the previous version. In this new version, changes in the manuscript are highlighted in yellow, so they are easy to identify. We have also included the changes made as a result of each comment as part of this document. Please note that our answers refer to the line numbers in the new version, corresponding to the requests of the reviewers.

We used blue highlight for reviwer 1

We used yellow highlight for reviewer 3

We used red letters for editor

Thank you for reviewing our article and contributing with your feedback to generate an improved version of the paper, which we hope will satisfy the concerns you raised with the previous version. In this new version, changes in the manuscript are highlighted in yellow, so they are easy to identify. We have also included the changes made as a result of each comment as part of this document. Please note that our answers refer to the line numbers in the new version, corresponding to the requests of the reviewers.

Point 1:

A serious note: how is this paper different from your previously published paper on competencies (Gonzales et al 2020)? It seems to be considerable overlap between these two papers. Please explain how this paper is different and what unique contributions are found in this paper!

Response 1: Thank you very much for your appreciation.

In the previous article you mentioned, we addressed the core of competencies that are necessary for nurse managers in Spain. Our panel of experts agreed on a core of 8 competencies for each of the functional roles (operational, logistical and senior management) of nurse management. The expert panel also agreed on the development needed for each core competency and how to achieve this development.

In contrast, the research presented in this article consists in the first competency model for middle nurse managers validated in Spain. This is where the main strength and novelty of this article lies. Our panel of experts reached a consensus on a model for the nurse manager of 51 competencies, at the levels of development described. This article is key for middle nurse managers. The research allows the establishment of selective and training processes adapted to this competency model. This could mean a radical change for middle nurse management in Spain.

Point 2:

Methods: please elaborate on the different levels of development. Are the levels mutually exclusive or are they cumulative? How are we to understand the content of the different levels? Please elaborate!

Response 2: Thank you very much for your appreciation, in lines 165-194 we have included your request  and we have rewritten this section

Lines 175 – 176. We've included it in the manuscript:

The degree of development must be understood in a progressive and exclusive manner: starting on the degree of novice and ending at the level of expert. The degree of development is achieved through the appropriate training to each level.

On the other hand, similar degrees of development are reported in the literature, for example:

Chase, L. (2010). Nurse manager competencies. [University of Iowa]. https://doi.org/10.1097/00006247-199403000-00008

Player, K. N., & Burns, S. (2015). Leadership Skills: New Nurse to Nurse Executive. Nurse Leader, 13(6), 40–43. https://doi.org/10.1016/j.mnl.2015.09.008

Thomas, J., & Herrin, D. (2008). The executive master of science in nursing program: competencies and learning experiences. The Journal of Nursing Administration, 38(1), 4–7. https://doi.org/10.1097/01.NNA.0000295635.26087.31

Thank you very much for such an important appreciation.

Point 3:

Discussion: Make an overview of the discussion section and see if you can omit certain sentences and expressions. For instance, you do not need to refer back to the rounds of the Delphi method in the second paragraph. Suggestion: start each paragraph with repeating the specific objective of the paper.

Response 3: Thank you very much for your appreciation. We have taken into account your comments and corrected some parts of the manuscript.

Point 4:

Line 31: Delete unnecessary adjective “enormous”.

Response 4: Thank you very much for your appreciation. It is corrected in the line 32.

Point 5:

Line 95-101: Suggestion, enumerate the specific objectives of the paper.

Response 5: Thank you very much for your appreciation. We have reviewed the objectives

Point 6:

Line 104: Please add a reference to the term scoping review.

Response 5: Thank you very much for your appreciation. We added in line 109:

Arksey, H., & O’Malley, L. (2005). Scoping studies: Towards a methodological framework. International Journal of Social Research Methodology: Theory and Practice, 8(1), 19–32. https://doi.org/10.1080/1364557032000119616

Point 7:

Line 135,140,145,147,155: Delete the bullets for readability.

Response 7: Thank you very much for your appreciation. We deleted all of bullets as you can see between lines 132-152. In this case we have not highlighted the text to reduce confusion

Point 8:

Table 1: change the interval 41-50 to 40-50 for age.

Response 8: Thank you very much for your appreciation. We changed table 1

Point 9:

Table 2: Check the names of the skills (e.g., “Estrategic”, “Lidership”). Categories V and VI should also be in bold face.

Response 9: Thank you very much for your appreciation. We have corrected it.

Point 10:

Table 4: “Lidership”.

Response 10: Thank you very much for your appreciation. We have corrected all mistakes.

Point 11:

Line 222: Ad a heading for the Discussion section.

Response 11: Thank you very much for your appreciation. We have corrected.

Thank you so much for reviewing our paper

Your feedback has undoubtedly improved our paper.